# Inspection and Assessment of Corrosion in Pretensioned Concrete Bridge Girders Exposed to Coastal Climate

**Magdalena J. Osmolska** [1,2,*] **, Karla Hornbostel** [3] **, Terje Kanstad** [2] **, Max A.N. Hendriks** [2,4] **and Gro Markeset** [1]

1   Department of Civil Engineering and Energy Technology, Oslo Metropolitan
    University (OsloMet—Storbyuniversitetet), 0166 Oslo, Norway; gromark@oslomet.no
2   Department of Structural Engineering, Norwegian University of Sciences and Technology (NTNU),
    7491 Trondheim, Norway; terje.kanstad@ntnu.no (T.K.); max.hendriks@ntnu.no (M.A.N.H.)
3   Norwegian Public Roads Administration (NPRA), Directorate of Public Roads, 7030 Trondheim, Norway;
    karla.hornbostel@vegvesen.no
4   Faculty of Civil Engineering and Geosciences, Delft University of Technology (TU Delft),
    2628 CN Delft, The Netherlands
*   Correspondence: magdap@oslomet.no; Tel.: +47-939-57-670

**Abstract:** The most common methods for detecting chloride-induced corrosion in concrete bridges are half-cell potential (HCP) mapping, electrical resistivity (ER) measurements, and chloride concentration testing, combined with visual inspection and cover measurements. However, studies on corrosion detection in pretensioned structures are rare. To investigate the applicability and accuracy of the above methods for corrosion detection in pretensioned bridge girders, we measured pretensioned I-shaped girders exposed to the Norwegian coastal climate for 33 years. We found that, even combined, the above methods can only reliably identify general areas with various probabilities of corrosion. Despite severe concrete cracking and high chloride content, only small corrosion spots were found in strands. Because HCP cannot distinguish corrosion probability in the closely spaced strands from other electrically connected bars, the actual condition of individual strands can be found only when concrete cover is locally removed. Wet concrete with high chloride content and accordingly low HCP and low ER was found only in or near the girder support zones, which can therefore be considered the areas most susceptible to chloride-induced corrosion. We conclude by proposing a procedure for the inspection and assessment of pretensioned girders in a marine environment.

**Keywords:** pretensioned girders; chloride-induced corrosion; inspection; half-cell potentials; electrical resistivity; chloride concentrations; corrosion assessment

## 1. Introduction

Chloride-induced corrosion in aging pretensioned concrete bridge girders due to exposure to an aggressive coastal environment, or inadequate durability design, is now recognized as an increasing deterioration problem in coastal bridges in Norway and globally [1–3]. A study of 227 pretensioned girder bridges exposed to the Norwegian coastal climate (inner coastal, for example inner fiords; coastal; and harsh coastal with extreme coastal weathered conditions) revealed that about 37% have reinforcement corrosion in pretensioned girders because they have less concrete cover than the minimum that was required by Norwegian regulations when they were built [2]. The most severe chloride-induced corrosion damage was observed on the concrete surface of the inner girders (typically

the second and third girders from the windward side) in or near the support zones. The location of the corrosion damage was explained by the interaction between geometry and environmental exposure [2].

Chloride-induced corrosion leads to localized pitting, which reduces the reinforcement cross-section, yield strength, and ductility. In pretensioned girders, the concrete section is subjected to compressive stresses (introduced by pretensioning), which limits the visible cracking caused normally by an expansive corrosion product (rust). In addition, strands experience high levels of stresses during the bridge's service life, which increases the rate and likelihood of corrosion. Pitting corrosion can result in the fracture of highly stressed strands and ordinary reinforcement, which may considerably reduce the load-bearing capacity of the girders. Consequently, reliable inspection methods are crucial for the detection of corrosion before serious damage occurs.

The strands in pretensioned girders are placed in concrete without ducts, which means that conventional methods developed for ordinary reinforced concrete (RC) can be used to detect reinforcement corrosion. The most common non-destructive (NDT) in situ tests for concrete bridge inspection are visual inspection, concrete cover mapping, concrete electrical resistivity (ER) measurements, and half-cell potential (HCP) mapping. One common destructive test (DT) is chloride concentration measurement. However, each of these tests have their limitations and the interpretation of tests results is not always straightforward.

The main objective of this research was to determine a reliable procedure for detecting corrosion in pretensioned concrete bridge girders, which combined HCP mapping, ER measurements, chloride content testing, and concrete cover measurements, with visual inspection. In addition, it aimed to describe the distribution of corrosion probability along pretensioned concrete girders and analyze the factors influencing their corrosion in a marine environment.

To investigate the applicability and accuracy of the above measurement methods for corrosion detection in pretensioned bridge girders, we carried out a study based on experimental data collected during field investigations of Dalselv Bridge, a 33-year-old girder bridge exposed to the Norwegian coastal climate.

## 2. Theoretical Background

### 2.1. Concrete Cover and Critical Chloride Content

Chloride ingress in RC structures is mainly governed by the quality (permeability) of the concrete, the thickness of the concrete cover to the reinforcement, and the level of chloride the structure is exposed to. Depassivation and the onset of corrosion may occur when the chloride concentration at the reinforcement surface reaches a critical level. The concrete cover thickness and the chloride content level at the reinforcement surface are therefore crucial parameters for the likelihood of corrosion initiation.

The probability of corrosion can be assessed by comparing the chloride concentration obtained at the reinforcement surface with the statistical distribution of critical chloride content or threshold for corrosion initiation. A probabilistic approach is needed due to the large scatter of chloride threshold values reported in the literature for both ordinary [4–8] and prestressing [9–14] steel. The fib Model Code for Service Life Design [15] suggests a beta distribution for the critical chloride content for reinforcement depassivation, with a lower bound of 0.2% and a mean value of 0.6% ± 0.15% by weight of cement. Markeset investigated field data from Norwegian quays and proposed a log-normal distribution for critical chloride content, with a higher mean value of 0.77% by weight of cement and a coefficient of variation (COV) of 32% [16].

Despite the stochastic nature of the chloride threshold, the conservative value of 0.4% by weight of cement recommended by CEB [17] is often used for RC assessment. For prestressed concrete, however, CEB recommends a lower chloride threshold of 0.2% by weight of cement [17] because prestressing steel is more sensitive to corrosion in terms of stress corrosion cracking and hydrogen embrittlement [18]. These threshold values coincide with the limits given by European Standard EN-206 [19] for the maximum chloride content allowed in new structures.

Nevertheless, a study of a post-tensioned precast concrete girder from the now demolished Sorell Causeway Bridge in Australia [20] found severe corrosion of stirrups and adjacent post-tensioning tendons for lower chloride concentrations than the conservative thresholds of 0.2% and 0.4% by weight of cement. Furthermore, research on the box-girder Gimsøystraumen Bridge [21] in Norway found corrosion for a chloride content as low as 0.01% by weight of cement. These findings show that the CEB recommendations [17] should be treated with caution.

Chloride concentration testing of many densely spaced bars is quite challenging (especially the core-drilling part). The samples must usually be collected using the powder-drilling method, which results in large depth intervals on the chloride profile and consequently high uncertainties in actual chloride concentrations. Moreover, the cover for individual strands may not be detected due to equipment limitations (such as a minimum reinforcement-spacing to cover-thickness ratio).

## 2.2. Concrete Resistivity

Electrical resistivity (ER) quantifies how strongly a material resists the transport of current. In concrete, ER mainly depends on the moisture content, but it is also influenced by the concrete quality (cement type, water–cement (w/c) ratio), temperature, and chloride content [22]. In relatively homogeneous concrete, areas with high and low ER usually indicate dry and wet areas, respectively [23]. However, even for the same concrete mix and exposure conditions, ER has been found to have a significant scatter, and a 20–25% COV must be considered normal for ER measured in the field [23]. Furthermore, cracks and delamination in the concrete cover, which must be considered when evaluating ER results, especially in damaged areas, have been found to influence concrete resistivity [24–27].

Many researchers have proposed criteria for corrosion activity assessment (sometimes denoted as corrosion risk) by relating ER ranges to corrosion rates (negligible to very high) [28–32]. However, in a review of the literature, Hornbostel et al. [33] found a large scatter between ER-corrosion rates. RILEM TC 154 [23] suggests general criteria for ER assessment in relation to corrosion probability rather than rate; see Table 1. A literature review by Song and Saraswathy [34] confirms RILEM's ER boundary for high corrosion probability, but suggests a significantly lower ER boundary for negligible corrosion probability. The large scatter in ER ranges reported in relation to both probability and rate indicates that corrosion assessment based on ER alone may not give reliable results [35].

**Table 1.** Electrical resistivity in relation to corrosion probability based on RILEM TC 154 [23].

| Probability of Corrosion | Electrical Resistivity in $\Omega$m |
| --- | --- |
| Negligible | Higher than 1000 |
| Low | Between 500 and 1000 |
| Moderate | Between 100 and 500 |
| High | Less than 100 |

## 2.3. Half-Cell Potential Mapping

HCP measurements enable the identification of areas with varying probability of corrosion. ASTM C876 [36] proposes criteria based on absolute potential values for the interpretation of HCP measurements; see Table 2. These criteria, however, were developed for ordinary reinforcement in bridge decks exposed to de-icing salts [37] and their accuracy in other exposure conditions and/or other reinforcement configurations/types (e.g., pretensioned reinforcement) is uncertain. A large number of parameters can influence the numerical values of the potentials measured, including concrete cover thickness, moisture content, availability of oxygen, w/c ratio, chloride content and carbonation depth, the presence of cracks and delamination, the ambient temperature, and the pre-wetting time [38–41]. RILEM TC 154-EMC [38] therefore recommends analyzing HCP maps to detect localized minima and steep gradients. As a complementary approach, RILEM also suggests statistical analysis of the HCP data [38], which can be performed based on the procedure developed by Gulikers and Elsener [42].

Most studies in the literature describe HCP measurements performed on ordinary RC structures, and very few are related to pretensioned elements [40,43,44].

**Table 2.** Electrochemical potential in relation to corrosion probability based on ASTM C876 [36].

| Probability of Corrosion | Half-Cell Potential in mV CSE (Copper/Copper Sulphate Electrode) |
|---|---|
| Less than 10% (low) | More positive than −200 |
| Uncertain | Between −350 and −200 |
| More than 90% (high) | More negative than −350 |

## 2.4. Combining NDT Methods

Because HCP measurements are influenced by numerous factors, including moisture, corrosion assessment based on HCP results alone may lead to incorrect conclusions. In structures exposed to chlorides, concrete with high moisture content (low ER) provides an environment favorable for reinforcement corrosion (both for the ingress of chlorides and high corrosion rates). Consequently, the low ER will strongly influence the numerical values of HCP measurements. To obtain more reliable evaluation of factors influencing corrosion, RILEM recommends assessing HCP results in combination with ER measurements [38].

Sadowski [45] presents such a methodology. He divides the area measured into three corrosion probability ranges: (1) low HCP and ER indicating a corrosion probability of more than 90%; (2) high HCP and ER indicating a corrosion probability of less than 10%; (3) low HCP and high ER suggesting an uncertain corrosion state. However, this methodology uses the potential thresholds given in ASTM C876 [36] and a fixed ER threshold of 40 Ωm.

Because there are no absolute values for HCP and ER thresholds, Pailes [46] developed a statistically based approach to determine HCP and ER thresholds. Analyses of measurements from twelve bridge decks exposed to chloride ingress [46] found high corrosion probability for potentials below the range of −250 to −450 mV CSE (HCP thresholds for active corrosion). The ER threshold below which concrete provided a corrosive environment ranged between 350 and 530 Ωm. However, the applicability of these thresholds to pretensioned girders is uncertain.

A numerical study by Kessler and Gehlen [47] found that inhomogeneous moisture content below the concrete surface measured can reduce the detectability of reinforcement corrosion, and suggests evaluating HCP data only from parts of the structure with similar moisture conditions [48,49].

## 2.5. Combining NDT with DT Methods

One possible approach is to supplement HCP and ER measurements with chloride content measurements. By correlating HCP results with the chloride content, the chloride distribution and threshold can be roughly estimated for a structure [22]. However, corrosion assessment in RC structures should always be verified through local visual reinforcement inspections [38,50], which require spot-wise removal of the concrete cover.

Moreover, the previously mentioned research on the Gimsøystraumen Bridge [21] showed that chloride and HCP thresholds for corrosion are not single values, even for one structure. Instead, varying reinforcement corrosion states are associated with wide ranges of both chloride content and HCP. For example, no corrosion was found for a chloride content of 0.13 ± 0.13% by weight of cement and HCP of −49 ± 90 mV CSE, while small corrosion spots were observed for a chloride content of 0.46 ± 0.45% by weight of cement and HCP of −148 ± 111 mV CSE. The uncertainty in corrosion assessment decreases only when multiple measurements lead to similar conclusions.

## 2.6. Investigations of Corrosion in Pretensioned Girders

Novokshchenov [43] found a good correlation between HCP measured on strands in pretensioned concrete girders and corrosion damage detected during a visual inspection. Similarly, a study on

pretensioned girders of the Tiwai Point Bridge in the New Zealand coastal environment [51] revealed a strong correlation between low HCPs and areas of active strand corrosion. Nevertheless, the authors state that the HCP maps did not provide significantly more information than detailed visual inspection [51].

Nakamura et al. [40] performed a corrosion probability assessment of a pretensioned I-shaped bridge girder in Japan. Localized corrosion was found in web reinforcement only for the most negative potentials close to the support, where the chloride content was approximately 2.55 kg/m$^3$ (about 0.7% by weight of cement). Nevertheless, the corrosion potential detected was more positive than −350 mV CSE. The authors found that HCP assessment based on potential maps together with potential gradients was a reliable tool for detecting corrosion probability [40]. It should be noted that the condition of the strands was not visually verified in this study.

The statistical analysis of HCP measurements taken on adjacent pretensioned box girders from three bridges in Pennsylvania [44] showed that the probability of corrosion for potentials lower than −350 mV CSE was only 45%. Moreover, the HCP measurements were significantly scattered compared to the condition of the strands. For instance, strand corrosion was not observed for an average HCP of −197 mV CSE with a COV of 56.1%, while pitting was found for an HCP of −316 mV CSE with a COV of 32.9%. The research revealed the poor effectiveness of the HCP method for detecting strand corrosion. Based on the visual inspection and strand corrosion state, the authors found that the probability of detecting corrosion in the absence of cracks was only 10%, while in presence of cracks it was more than 70% [44].

In the above bridges [44], the average chloride concentration (from de-icing salts) found for non-corroded strands was 0.0113% by weight of concrete, which is lower than the ACI 318-08 [52] chloride threshold of 0.013% by weight of concrete (0.06% by weight of cement). Corrosion was detected for an average chloride content of 0.07% by weight of concrete. However, the authors reported a large variation in the chloride levels compared to strand conditions (no corrosion, light corrosion, pitting, and heavy pitting). For example, no corrosion was found for chloride levels of 0.082% by weight of concrete, while heavy pitting was observed for chloride levels as low as 0.0052% by weight of concrete [44].

The investigation on pretensioned NIB girders in the Hafrsfjord Bridge revealed that, for almost the same chloride content, corrosion may or may not occur [1,53]. The authors conclude that local differences in concrete quality might be the reason [1].

Due to the limited number of studies on the condition of pretensioned girders and discrepancies between the assessed and actual reinforcement corrosion reported in the literature, more research is needed to evaluate the accuracy of corrosion detection based on conventional methods.

## 3. Field Investigation of Dalselv Bridge

### 3.1. Bridge Details

Dalselv Bridge is a 40 m long, two-span bridge consisting of nine simply-supported standardized I-shaped precast pretensioned (NIB) girders in each span; see Figure 1. The bridge was built in 1985 in the inner coastal climate (fjord area) of the northern part of Norway. The superstructure is close to sea level (approximately 2.3–2.9 m measured at high and low tide), which implies high chloride loads on the bridge superstructure. According to data from the Norwegian Meteorological Institute collected in the years 2011–2019, the prevailing winds come from the sea side, from the south-west; see Figure 1a. The average temperature and relative humidity are 4.6 °C and 77%, respectively.

The NIB girders of Dalselv Bridge are made of concrete class C55 with a characteristic cube compressive strength (100 × 100 × 100 mm) $f_{ck, cube}$ = 55 MPa. The cylinder strength corresponds to about 80% of the cube strength. There were no requirements for a maximum w/c ratio when it was built, but such compressive strength would normally be achieved using a w/c ratio of about 0.40 [54]. The NIB girders have not been surface treated.

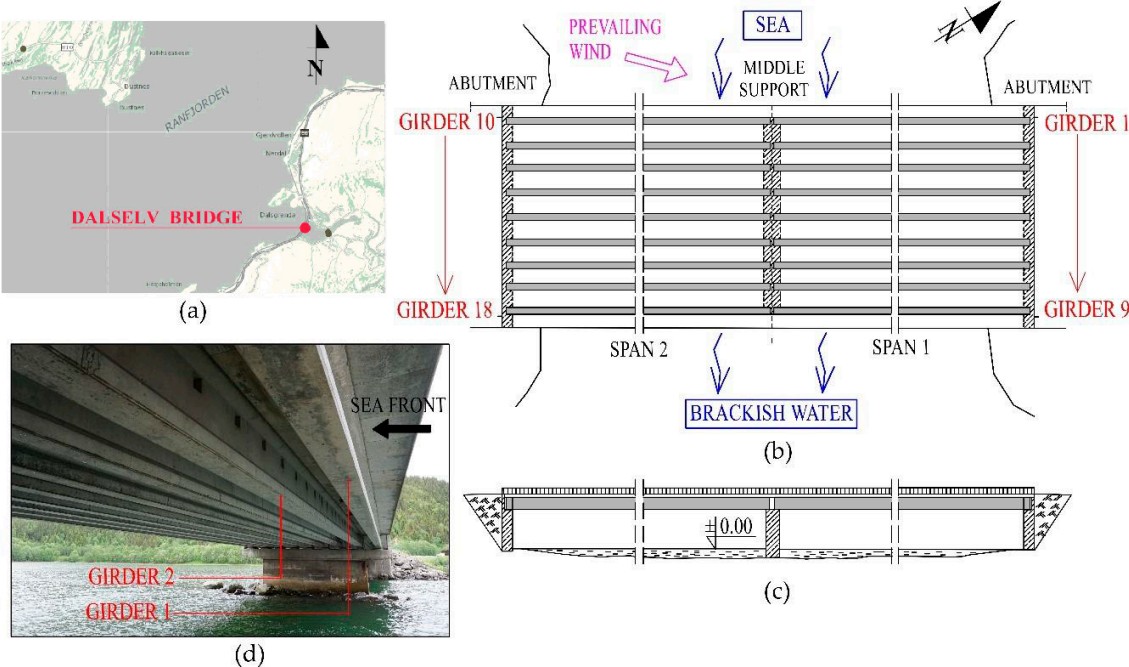

**Figure 1.** Dalselv Bridge: (**a**) bridge location; (**b**) bottom view; (**c**) side view; (**d**) photo taken underneath the bridge, facing south.

## 3.2. Inspection and Test Methods

The bridge was inspected in June 2018. The condition of the NIB girders was initially assessed based on a visual inspection, during which cracking, delamination, and wet concrete areas were documented. The second NIB girder facing the sea in the first span (Girder 2) was selected for more detailed investigation; see Figure 1.

The concrete cover to the stirrups was measured with a magnetic cover meter. The densely placed strands in the girder's bottom flange meant that the cover to the individual strands could not be detected with a cover meter, so the cover to the strands was measured on the end surface of the girder above the middle support, where the strands were visible. The corrosion probability in Girder 2 was assessed based on ER and HCP measurements, and combined with results from chloride sampling. To gather supplementary information, ER and chloride concentrations were measured locally in two other NIB girders. The reinforcement was inspected near two selected locations of chloride sampling, where concrete cover was locally removed.

Details of the experimental setup for measuring concrete resistivity, half-cell potentials, and chloride content measurements are presented in Sections 3.2.1 and 3.2.2.

### 3.2.1. Concrete Resistivity and Half-Cell Potentials

The concrete resistivity in Girder 2 was measured in shadow using a four-point Wenner Probe, in sunny weather with an air temperature of 16 °C. The concrete ER was measured on the side surface of the bottom flange (see Figure 2a) to avoid measurements on the pre-wetted bottom surface of the girder as recommended in RILEM TC 154 [23]. The ER measurements started at a distance of 1400 mm from the middle support and were then recorded approximately every 640 mm along the girder length, as indicated by the two ER points in Figure 2b. The ER was also measured locally in the web of Girder 2 at the location for chloride sampling, P3 (see Figure 2b), as well as on the bottom surface of Girders 1 and 11 about 1.5 m from the middle support.

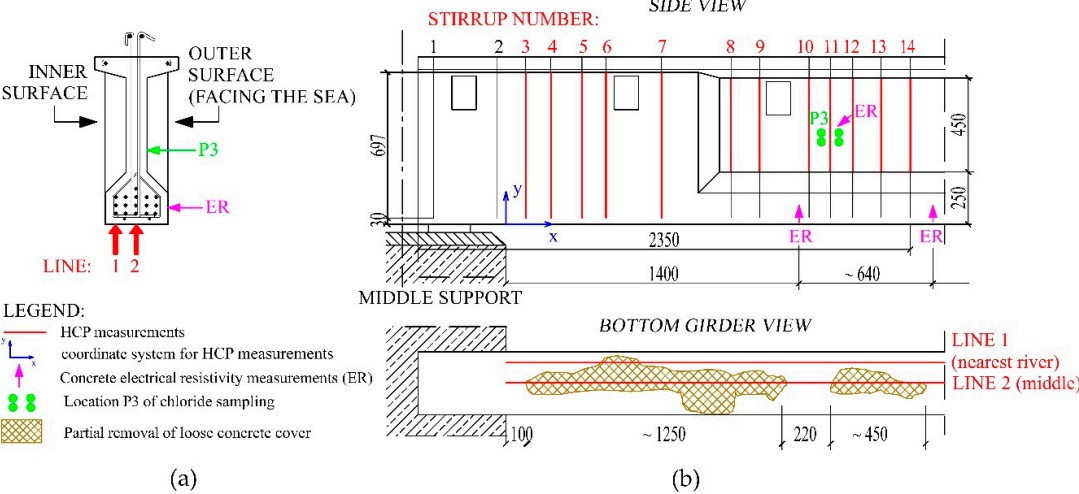

**Figure 2.** Location of half-cell potential (HCP) and electrical resistivity (ER) measurements in Girder 2: (**a**) measurements in the girder's bottom flange; (**b**) measurements (mm) in the web above the stirrups, and removal of loose parts of concrete cover.

To locate areas with high corrosion probability in Girder 2, HCP mapping was performed along the whole span length of the bottom surface of the girder, see Figure 2a. The measurements followed two lines to detect possible variations in corrosion probability between the strands. Moreover, to assess the condition of the vertical reinforcement in the shear zone near the middle support, the HCP was measured above the stirrups (numbered 3–14 in Figure 2b) on the inner and outer vertical surfaces of the girder.

Prior to the HCP measurements, the loose parts of the concrete cover (with a thickness of about 5 to 20 mm) were locally removed from the girder's bottom flange over a distance of approximately 2.1 m from the middle support, see Figure 2b. The HCP measurement procedure followed RILEM TC 154-EMC recommendations [38]. The electrical continuity of the reinforcement was verified at two points. The second point was used to establish an electrical connection between the reinforcement and the high-impedance voltmeter built into Profometer Corrosion (from Proceq). To reduce fluctuation in the values measured, the concrete surfaces were sprayed with tap water before the measurements. Half-cell potentials were measured against a copper/copper-sulphate electrode (CSE), using a one-wheel electrode type. To increase the probability of corrosion detection, the HCP values were recorded with a fine grid [48], every 50 mm along the lengths measured.

### 3.2.2. Chloride Concentrations and Reinforcement Inspection

Chloride concentrations in Girder 2 were measured at four locations, three of which were in the girder's bottom flange: P2 close to the middle support, P6 close to the abutment, and P5 in the middle of the span; see Figure 3. The fourth sampling point, P3, was located in the girder web; see Figure 2b. Additional chloride measurements were taken from the bottom flange of Girder 1 (P4) and Girder 11 (P1) at the same distance from the middle support as P2; see Figure 3a.

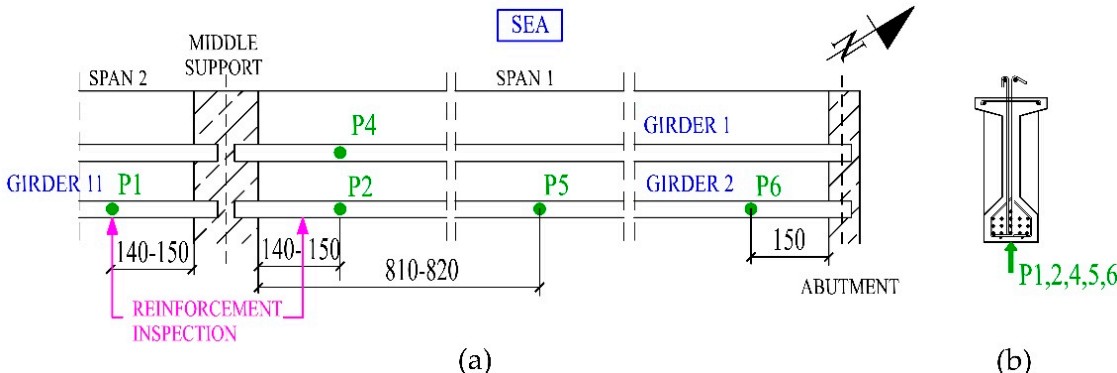

**Figure 3.** Locations P1, P2, P4, P5, and P6 for chloride sampling (distances in cm): (**a**) girder layout; (**b**) girder cross section.

The densely placed strands in the bottom flange of the pretensioned girders meant that samples for chloride analysis had to be obtained by drilling four holes per location with a 16 mm drill. Concrete powder samples were collected up to a depth of 70 mm in steps of 10 mm. Chloride concentrations in the dissolved powder samples were determined by the Norwegian research institute SINTEF using potentiometric titration. The results are presented as chloride profiles, with the chloride concentration given as a percentage of dry concrete weight.

The outermost reinforcement was visually inspected in Girder 2 close to the middle support; see Figures 2b and 3a. Reinforcement was also inspected in Girder 11; see Figure 3a. First, concrete cover was removed up to the stirrups level in Girder 2 over a distance of approximately 1.5 m from the middle support. Next, the cover was removed in Girder 11 up to strand level next to the location for chloride sampling, P1, for a distance of approximately 200 mm along the girder's bottom flange.

## 4. Results from the Field Investigation

### 4.1. Visual Observations and Cover Thickness

Corrosion was found in the pretensioned NIB girders of Dalselv Bridge, in and near the support zones. The most severe corrosion damage was found close to the middle support, in the second and third girders facing the sea (Girders 2, 11, 12), and in the bottom flanges of the outermost girders facing the sea (Girders 1, 10); see Figure 4. Less severe corrosion cracking was detected in the bottom flanges of girders close to the abutments. Girders 1 and 2 also had severe cracking with visibly corroding strands in the top flange next to the middle support.

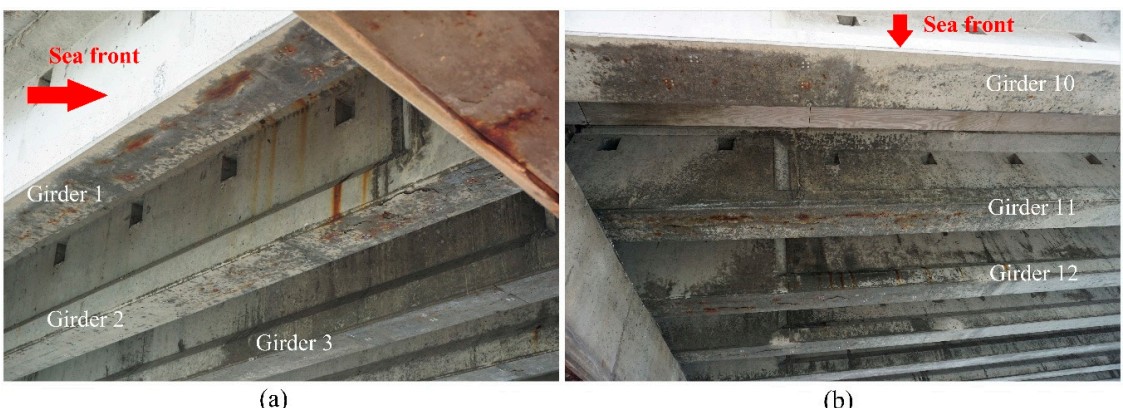

**Figure 4.** Corrosion-induced damage and humid areas in NIB girders: (**a**) in the first bridge span (1st to 3rd girders facing the sea) next to the middle support; (**b**) in the second bridge span (1st to 3rd girders facing the sea) next to the middle support.

At a certain distance from the middle support, the bottom surface of the first four NIB girders facing the sea had a significantly darker color. Moreover, the inner girders (Girders 2, 3, 11, 12, 13) had a darker color in the web next to the middle support; see Figure 4. We think this color change was due to a greater moisture load in these areas. The bottom flanges of the girders next to the abutments were also humid, although to a lesser extent than near the middle support.

The removal of the concrete cover in the bottom flange of Girder 2 revealed severe corrosion of mounting bars; see Figure 5a. Similar damage was found in the exposed mounting bars in Girder 11. Inspection of chloride sampling location P1 (in Girder 11) also revealed corrosion in stirrups and one strand; see Figure 5b. In the ø6 stirrups, the severest corrosion had accumulated in and near their bends, with a loss of diameter measured at about 50%. The horizontal part of the ø12 stirrup shows fairly moderate corrosion.

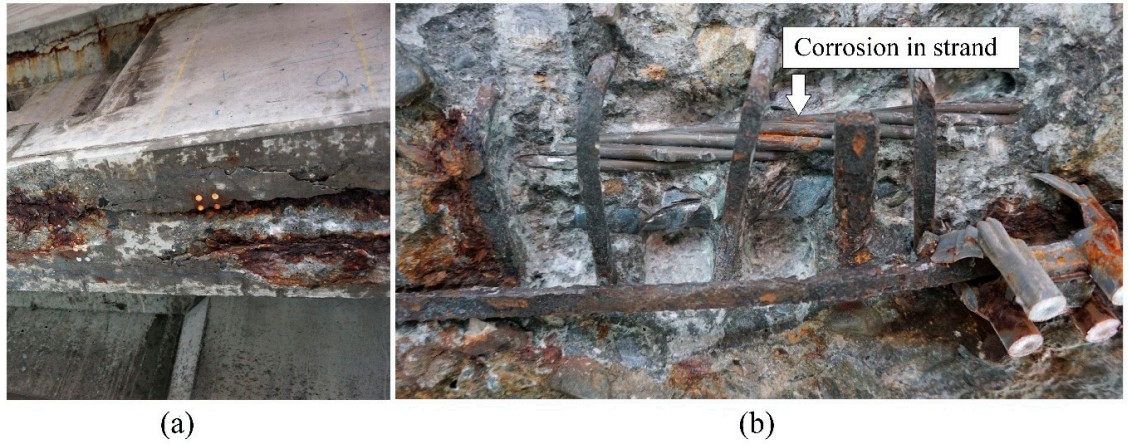

(a)                                                                                      (b)

**Figure 5.** Reinforcement corrosion: (**a**) next to chloride sampling location P2 in Girder 2; (**b**) next to chloride sampling location P1 in Girder 11.

Surprisingly, only small corrosion spots were found on the strand itself (see Figure 5), which suggests that corrosion started recently or that the corrosion rate is low. It should be noted that only a short length of strand was exposed and more corrosion spots could probably be found outside this area.

Measurement of the concrete cover up to the stirrups in Girder 2 revealed 5.5–7.5 mm more cover on the outer than the inner surface of the web, see Figure 6. This means that the mold must have moved before casting. During inspection, we also detected two visible mounting bars supporting stirrups in the girder's bottom flange, with plastic reinforcement chairs supporting the bars.

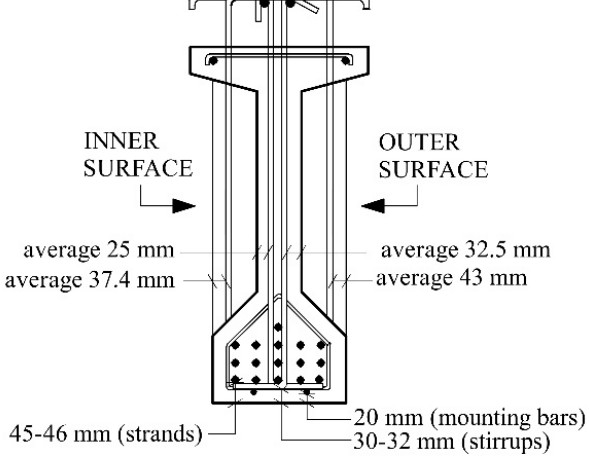

**Figure 6.** Concrete cover thickness measured in Girder 2.

### 4.2. Concrete Resistivity

The ER readings scatter significantly along the girder span from 230 Ωm to more than 1000 Ωm; see Figure 7. Overall, the ER measurements were rather high, which can indicate a good concrete quality (low w/c) with low concrete porosity.

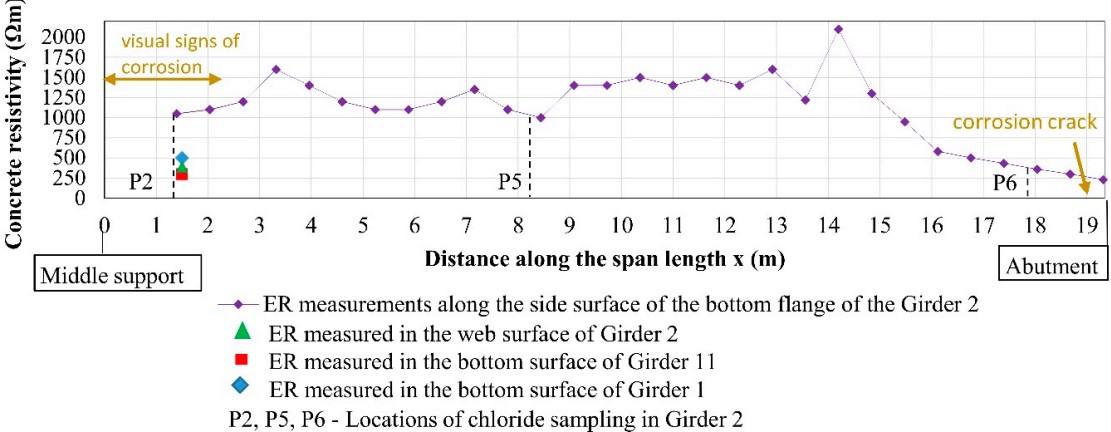

**Figure 7.** Concrete resistivity measured along the side surface of the bottom flange of Girder 2, in the web of Girder 2, and in the bottom surface of Girders 1 and 11 (about 1.5 m from the middle support).

For most of the span, the ER was greater than 1000 Ωm, which suggests dry concrete [23]. The ER readings decreased in the last approximately 4.5 meters to the abutment, which suggests a gradual change in moisture conditions; see Figure 7. Although wet areas observed during visual inspection suggested ER would also decrease near the middle support, high ER readings were recorded (more than 1000 Ωm). For comparison, the ER measurements in wet areas of Girders 1 and 11 close to the middle support were considerably lower (500 and 320 Ωm, respectively) than those in Girder 2; see Figure 7. In addition, the ER of 380 Ωm measured in the web of Girder 2 indicates high moisture next to the middle support. The high ER in the bottom flange of Girder 2 near the middle support may be due to delamination of the concrete cover caused by corrosion, the signs of which were visible over a distance of about 2.1 m from the middle support; see Figure 4a. Wind conditions and the location of the measurements on the side of the girder flange could also have had an impact.

### 4.3. Half-Cell Potentials

The HCP mapping along the bottom flange of Girder 2 is shown in Figure 8. Lines 1 and 2 both showed positive and stable HCP values (0 to 72 mV CSE) in the middle part of the span between 5 and 18 m from the middle support. Near the girder supports, HCP measurements decreased to their most negative values. The potential gradients near both supports were greater than 190 mV CSE/m, while no potential drops occurred in the middle of the span; see Figure 8. This indicates a higher probability of corrosion in reinforcement near the supports and a low probability of corrosion in the main part of the span.

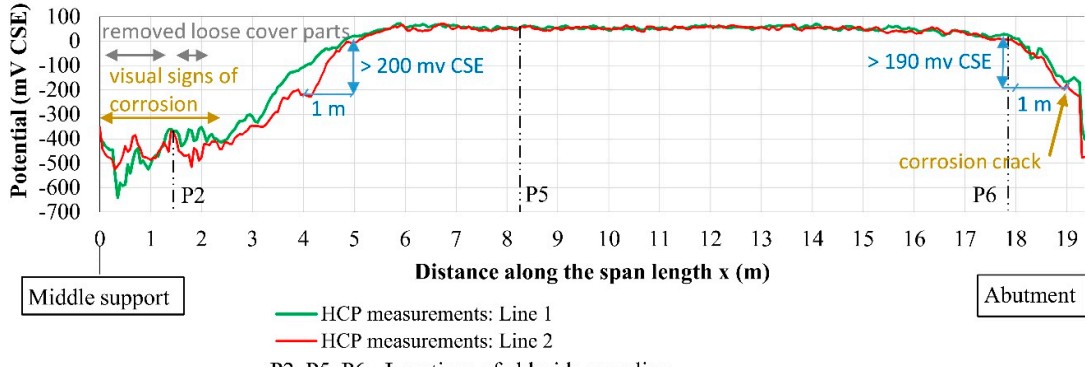

**Figure 8.** Distribution of half-cell potentials along the bottom surface of Girder 2.

The local scatter in HCP readings from 0.1 to 1.3 m and from 1.6 to 2.1 m from the middle support is probably due to partially removed loose concrete cover; see Figure 8. Outside the area of removed cover, the HCP readings in Line 2 are slightly lower than in Line 1; see Figure 8. Otherwise, the potential gradients in both lines show the same trend. In this case, therefore, it was not possible to detect variation in corrosion probability between closely spaced (about 50 mm) and electrically connected strands. This can be explained by either active corrosion in both strands or polarization of neighboring strands by a locally active corrosion area. The area polarized depends, amongst other things, on the ER. As the ER decreases, the area polarized increases.

The potentials measured above the stirrups near the middle support (numbered 3–14 in Figure 2b) are shown in Figure 9. The highest HCP values were measured in the upper corner of the girder end cross section, referred to here as the thick web. The potential distribution follows the moisture pattern observed in inner NIB girders close to the middle support (see Figure 4) and confirms the influence of moisture on HCP measurements.

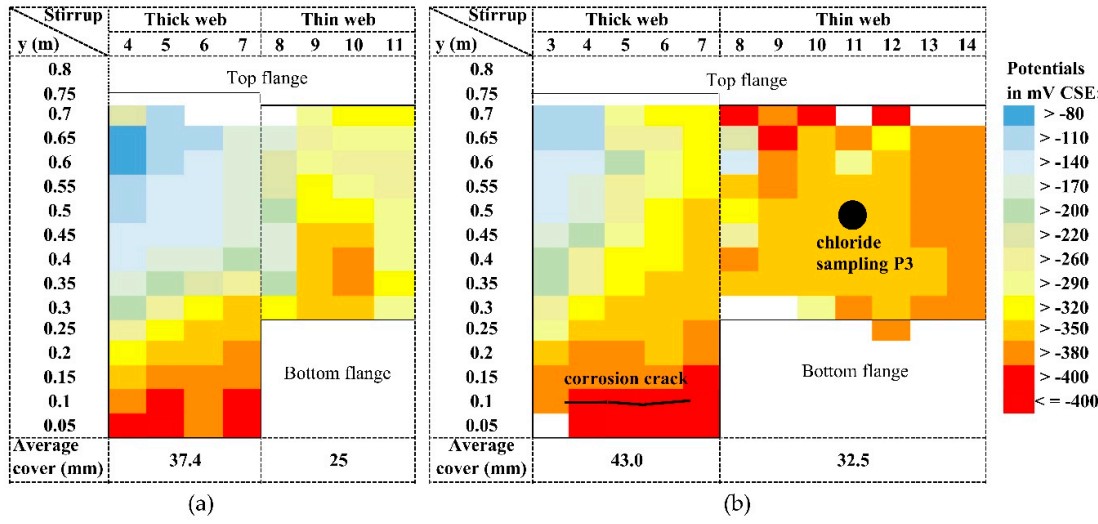

**Figure 9.** HCP measured above the stirrups in Girder 2: (**a**) inner surface; (**b**) outer surface (facing the sea) cf. Figures 2 and 6.

The most negative potentials (less than −380 mV) were found for parts of the stirrups at the bottom of the thick web. These locations correspond to the areas with visible corrosion damage shown in Figure 4a. The HCP readings above the stirrups show more negative values for the outer web surface (facing the sea) than for the inner surface. Despite its thicker cover (by 5.5–7.5 mm), the outer surface facing the sea is more likely to corrode than the inner surface.

### 4.4. Chloride Concentrations

The chloride profiles obtained for all of the sampling locations are presented in Figure 10. The chloride sampling locations on the bottom flanges of Girders 1, 2, and 11 (P1, P2, P4, P5, and P6) are given in Figure 3, and the chloride sampling in the web of Girder 2 near the middle support (P3) is given in Figure 2.

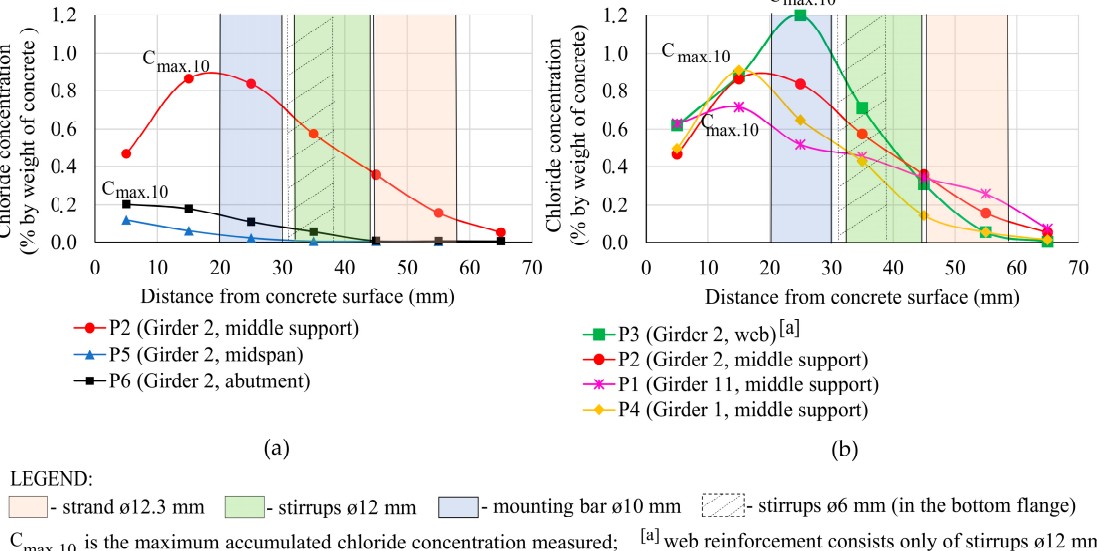

**Figure 10.** Chloride profiles obtained in Girders 1, 2, and 11: (**a**) along the bottom flange in Girder 2; (**b**) near the middle support in Girders 1, 2 and 11.

As shown in Figure 10a for the bottom flange of Girder 2, the highest chloride content at the level of the reinforcement was found near the middle support (P2). The chloride content in unsaturated concrete is strongly affected by the accumulation of surface chlorides [55]. The maximum accumulated chloride concentration, based on measurements at 10 mm concrete depth intervals (denoted $C_{max.10}$), was more than seven times higher near the middle support and almost two times higher near the abutment than in the midspan. This resulted in deeper chloride penetrations in the support zones of Girder 2.

The $C_{max.10}$ values are very high in all sampling locations near the middle support; see Figure 10b. This suggests comparable conditions for chloride ingress in the middle support vicinity.

The $C_{max.10}$ values obtained from locations P1–P4 are approximately 15–25 mm from the concrete surface (see Figure 10b), which indicates the presence of a convection zone, $\Delta x$. Although the sampling method used for obtaining chloride profiles does not allow us to accurately determine the depth of the convection zone due to the relatively large intervals between sampling points (10 mm), the $\Delta x$ seems to be greater near the middle support than in the midspan (P5); see Figure 10a.

The transport of chlorides behind the convection zone is governed mainly by diffusion, and it should be comparable for the given concrete composition and age. However, the chloride distributions in P1–P4 show some variations within the diffusion zone. P1 had the lowest $C_{max.10}$, but the highest chloride concentration at a depth of 50–60 mm, see Figure 10. This could be due to concrete inhomogeneity and internal cracks, which help moisture (containing chlorides) penetrate deeper into the concrete.

To convert chloride concentrations from percentage of concrete by weight to percentage of cement by weight, we assumed that 465 kg/m$^3$ of cement was used for the production of concrete C55 [56], and that the concrete density was 2450 kg/m$^3$ (as for the modern concrete class C45/55 with a compressive strength similar to C55).

Considering the conservative chloride thresholds of 0.2% and 0.4% by weight of cement for prestressed and ordinary reinforcement, respectively [17], corrosion was likely to occur in all reinforcement types in the locations tested near the middle support (P1–P4) and in mounting bars near the abutment (P6); see Table 3.

**Table 3.** Chloride concentrations at the level of reinforcement, in wt% of concrete/wt% of cement

| Sampling Location | Strands | Stirrups | Mounting Bars |
|---|---|---|---|
| P1 (Girder 11, middle support) | 0.344 [1]/1.81 | 0.48 [2]/2.53 | 0.62 [2]/3.27 |
| P2 (Girder 2, middle support) | 0.358 [1]/1.89 | 0.68 [2]/3.58 | 0.85 [2]/4.48 |
| P3 (Girder 2, web) | - | 0.81 [2]/4.27 | - |
| P4 (Girder 1, middle support) | 0.143 [1]/0.75 | 0.52 [2]/2.74 | 0.78 [2]/4.11 |
| P5 (Girder 2, midspan) | 0.005 [1]/0.03 | 0.01 [2]/0.05 | 0.04 [2]/0.20 |
| P6 (Girder 2, abutment) | 0.007 [1]/0.04 | 0.07 [2]/0.37 | 0.14 [2]/0.74 |

[1] Values measured; [2] Values interpolated based on chloride profiles. Values marked with red color exceed CEB [17] critical chloride concentrations of 0.2% and 0.4% by weight of cement for prestressed and ordinary reinforcement respectively.

## 5. Discussion

### 5.1. Environmental Exposure along the Girder

In the midspan of Girder 2 (P5), the maximum chloride content $C_{max.10}$ was the lowest of all of the locations tested; see Figure 10. This is probably due to the concrete being drier, which was also indicated by the high ER measurements in this area; see Figure 7.

High $C_{max.10}$ values and the presence of a convection zone were detected in all locations tested near the middle support; see Figure 10. Previous studies have shown that a convection zone is a typical phenomenon not only for structures in tidal and splash zones [15,57], but also for structures exposed to cyclic drying–wetting in a salt fog environment [58], i.e., a marine spray zone. Exposure to cyclic drying–wetting near the middle support would result in periodically high moisture content, which was indicated by the low ER in the web of Girder 2 and the bottom flanges of Girders 1 and 11, and by the wet concrete surfaces we observed; see Sections 4.1 and 4.2. There was no clear indication of a convection zone in the concrete near the abutment (see sampling location P6 in Figure 10a). $C_{max.10}$ was also significantly lower here than near the middle support. Nevertheless, the low ER measured near the abutment suggests high moisture; see Figure 7.

We can conclude that the exposure to moisture and chlorides varies along the girder length. In contrast to the dry conditions in the midspan, both support zones and their vicinity are probably exposed to substantial wetting and drying.

The differences in $C_{max.10}$ between locations tested near the abutment and the middle support may arise from varying frequency and length of wetting–drying cycles [57] and other differences in the micro-climate around the girders. Moreover, cover cracking due to corrosion of mounting bars in girder bottom flanges near the middle support (see Figure 5a) could have enabled more chloride ingress than the sound concrete at P6 near the abutment.

Higher maximum accumulated chloride concentrations close to supports were also observed in the box-girder Gimsøystraumen Bridge in Norway after 11 years of exposure to coastal climate [59]. Figure 11 shows that the $C_{max.10}$ next to the abutment (axis 1) was lower than that around the middle support (axis 2). The moisture content measured also varied along the box girder, with the highest values next to supports and lowest in the midspans [59], which confirms the findings in this study.

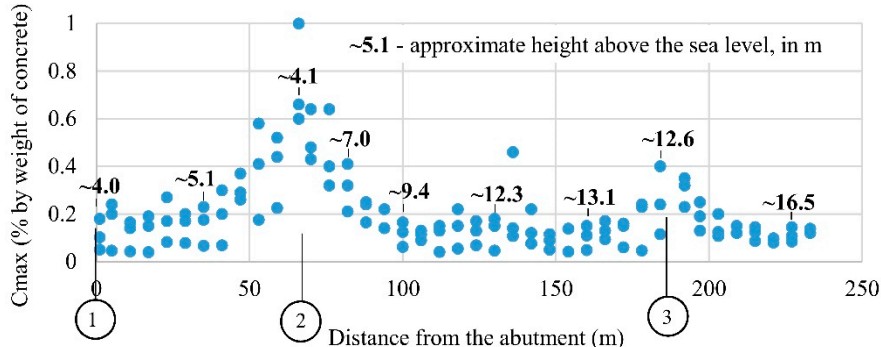

**Figure 11.** Chloride concentrations $C_{max}$ in Gimsøystraumen Bridge measured in a 10 mm thick concrete layer after 11 years of exposure to Norwegian coastal climate [59].

*5.2. Assessment of Corrosion Probability*

### 5.2.1. Based on HCP and ER Measurements

To find the probability of corrosion, statistical analyses of the HCP readings were performed following the procedure developed by Gulikers and Elsener [42]. The procedure divides the measured potentials into bins of 10 mV. Next, two Gaussian distributions representing active and passive potentials are fitted to the measurement data by regression analysis of its frequency density distribution [42], using the Maximum Likelihood Estimation method for estimating parameters for both distributions [60]. For each measured potential, *E*, the probability of corrosion $P_{act}(E)$ is calculated from Equation (1), where $p_{act}$ and $p_{pas}$ denote the probability density distribution of active and passive potentials, respectively.

$$P_{act}(E) = \frac{p_{act}(E)}{p_{act}(E) + p_{pas}(E)}$$

(1)

First, we assumed that Girder 2 is a homogenous structure with comparable exposure to chlorides and moisture. All 987 measured potentials were therefore considered in the statistical analysis. Results of regression analysis are presented in Figure 12.

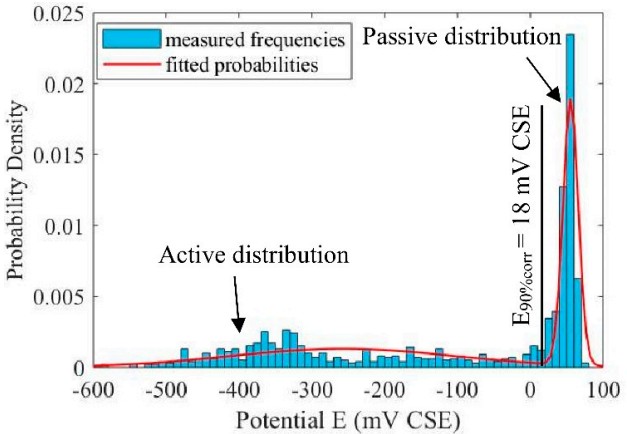

Parameters of the probability density distribution for passive potentials:
$\mu_{pas}$ ($E_{pas}$) = 55.36 mV
$\sigma_{pas}$ ($E_{pas}$) = 10.42 mV

Parameters of the probability density distribution for active potentials:
$\mu_{act}$ ($E_{act}$) = -257.61 mV
$\sigma_{act}$ ($E_{act}$) = 154.66 mV

Relative number of potentials for active corrosion:
$R_{act} = n_{act} / 987 = 0.509$

**Figure 12.** Frequency density distribution of measured and fitted potentials assuming homogenous environmental exposure in Girder 2. Parameters of the potential distributions.

A 90% probability of corrosion was found for a potential of 18 mV CSE, as indicated in Figure 12, while a corrosion probability close to 100% was calculated for potentials below 10 mV CSE. We found these values unrealistically high compared to commonly reported thresholds [40,46]. Moreover, no steep

potential gradients indicating corrosion were observed in the girder area with measured potentials greater than 0 mV CSE; see Figure 8.

To improve the assessment, statistical analysis should be performed on areas with similar environmental exposure to moisture and chlorides [48,49]. Since these conditions vary along NIB girders (see Section 5.1), Girder 2 was subdivided into two regions with comparable chloride and moisture loads: 1) a dry area in the span and 2) humid areas near supports. For this purpose, the relationship between ER and HCP was analyzed. We assumed that moisture conditions on the bottom and side surfaces of the girder flange were comparable. The potentials at ER measurement locations were calculated as the average values from Lines 1 and 2 over a distance of 200 mm.

As shown in Figure 13, we found a strong correlation between HCP and ER measurements for the right half of Girder 2. The potential decreases when the ER is below approximately 580–950 Ωm. These values are close to the ER thresholds of 500 and 1000 Ωm, above which the corrosion probability is low and negligible, respectively [23]. However, they are higher than the ER of 350–530 Ωm found by Pailes [46], which may be because NIB girders require a higher quality concrete than bridge decks. For the left half of Girder 2, no clear relationship was found between HCP and ER, see Figure 13. Despite a high ER, decreasing HC potentials with decreasing distance to the middle support showed the same trend as in the right half near the abutment; see Figure 8. We therefore concluded that HCP measurements were more reliable for corrosion assessment then ER results, which should always be verified with visual inspection.

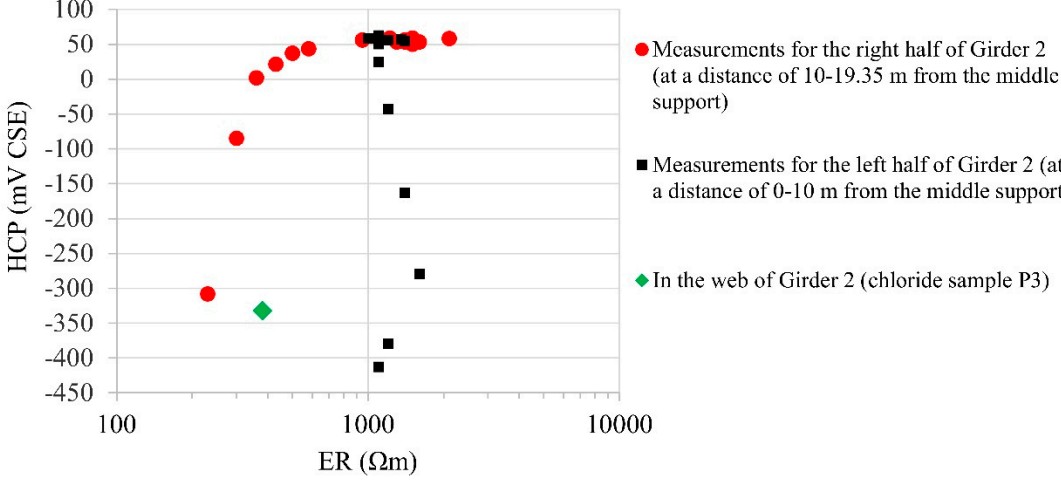

**Figure 13.** Relationships between HCP and ER measurements.

The exact ER threshold that separates zones with dry (non-corrosive) and humid (favorable for corrosion) environmental conditions could not be clearly specified for Girder 2. However, assuming the ER threshold lower than 980 Ωm, the region with elevated moisture was found to extend a distance of 3.8 m from the abutment; see Figure 7. Because low HCP measurements were found for a distance of about 6–6.5 m from the middle support (see Figure 8), the region near the middle support was initially estimated to extend 6.5 m. The results of regression analysis performed on the 642 potentials collected from these areas are shown in Figure 14. The calculated potential threshold for probability of corrosion greater than 90% was equal to −21 mV CSE.

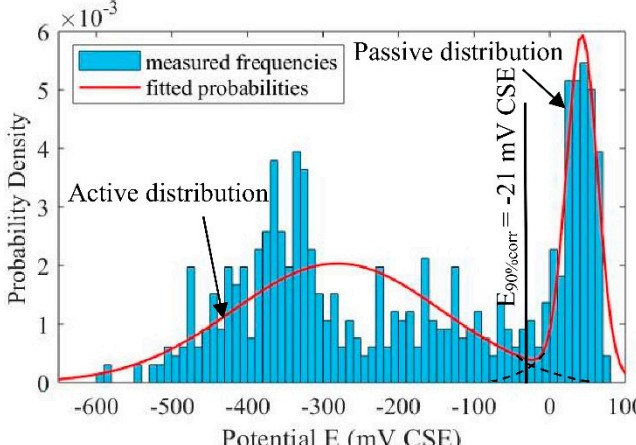

Parameters of the probability density distribution for passive potentials:
$\mu_{pas}$ ($E_{pas}$) = 39.74 mV
$\sigma_{pas}$ ($E_{pas}$) = 19.18 mV

Parameters of the probability density distribution for active potentials:
$\mu_{act}$ ($E_{act}$) = -281.12 mV
$\sigma_{act}$ ($E_{act}$) = 137.75 mV

Relative number of potentials for active corrosion:
$R_{act}$ = $n_{act}$ / 662 = 0.726

**Figure 14.** Frequency density distribution of measured and fitted potentials assuming similar moisture conditions near supports. Parameters of the potential distributions.

An additional analysis was performed for the region of increased moisture limited to the distance of 3.3 m from the abutment (ER lower than 580 Ωm) and 6 m from the middle support. In this case, the corrosion probability of 90% was calculated for a potential of −26 mV CSE. The calculated values of −21 to −26 mV CSE are higher than the potentials of −670 to −360 mV CSE measured in corroding areas, but slightly lower than the threshold of 18 mV CSE found from analysis of all of the data collected. The estimated corrosion probability can therefore be considered a conservative indicator of corrosion, and HCP measurements could be limited to areas with high levels of moisture in concrete. In contrast, statistical analyses that disregard varying exposure conditions in some cases may lead to overestimation of the area with high corrosion probability, and therefore additional but unnecessary repairs.

The calculated threshold is more positive than commonly proposed; e.g., Pailes suggested potentials of −250 to −450 mV CSE indicate high corrosion likelihood in bridge decks [46]. These values should therefore not be used for NIB girder assessment in general. Like Nakamura et al. [40], we found high corrosion probability for potentials more positive than those recommended in ASTM C876 [36]; see Table 4.

**Table 4.** Comparison of electrochemical potential in relation to corrosion probability based on ASTM C876 [36] and statistical analysis in this study.

| Probability of Corrosion | Half-Cell Potential in mV CSE According to ASTM C876 [36] | Half-Cell Potential in mV CSE Obtained in This Study |
|---|---|---|
| Less than 10% (low) | More positive than −200 | More positive than about 10 |
| More than 90% (high) | More negative than −350 | More negative than about −25 |

The corrosion probability assessment based on the HCP criteria given in ASTM C876 [36] would underestimate the extent of the zone with high corrosion probability near supports. For example, HCP of −195 mV CSE next to the abutment would indicate only a low corrosion probability despite evident corrosion cracks. Furthermore, according to the ER criteria suggested by RILEM TC 154-EMC [23], low to moderate corrosion probability would be found only near the abutment, despite visible corrosion damage near the middle support. This shows the inconsequence of these numerical HCP and ER criteria for corrosion probability assessment.

5.2.2. Based on Chloride Test Results

To the authors' knowledge, the statistical distribution of chloride thresholds for pretensioning reinforcement has not yet been studied. We therefore assessed corrosion probability for both strands and ordinary reinforcement using the statistical model proposed by Markeset [16]. The model was developed for ordinary reinforcement bars based on field data from quay elements long exposed to the Norwegian marine environment and with w/c ratios that vary between 0.4 and 0.5.

Using this model, we found that the corrosion probability was close to 100% for strands and stirrups close to middle support in the inner girders (P1–P3), whereas strands in the midspan (P5) and abutment vicinity (P6) had a negligible corrosion probability of less than 1%; see Figure 15.

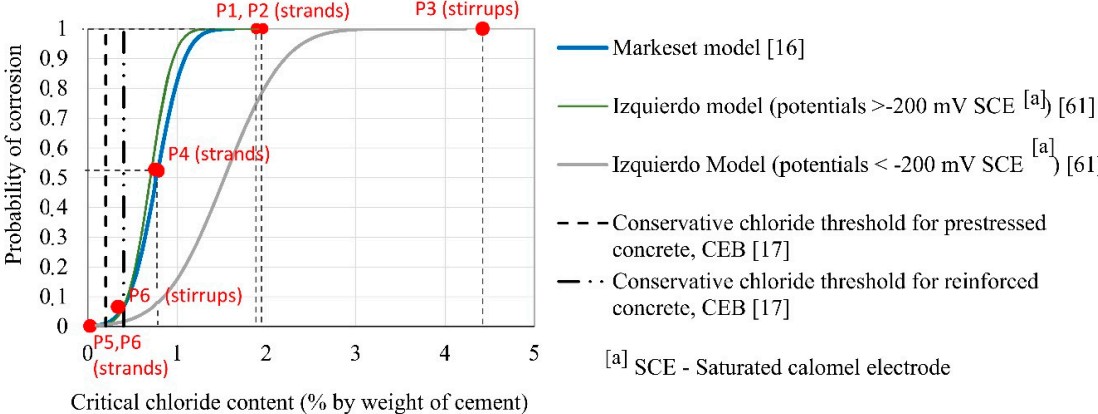

**Figure 15.** Probability of reinforcement corrosion calculated based on the Markeset statistical model.

Strands in the outermost girder (P4) had a lower probability of corrosion than strands in the inner girders we tested. Since the severest damage observed was in inner girders facing the sea, these girders are probably critical for corrosion assessment.

It should be mentioned that such evaluated corrosion probabilities may differ when other statistical models are used. Moreover, the chloride threshold distribution strongly depends on the potential of the steel. For potentials higher than $-200$ mV SCE ($-128$ mV CSE), Izquierdo's statistical model [61] is very similar to the Markeset model [16], but for potentials below $-200$ mV SCE, Izquierdo projects higher thresholds; see Figure 15. The same is true of Pedeferri's diagram [22]. Low (more cathodic) steel potentials can be due to wet concrete or polarization from neighboring corroding bars. In pretensioned girders, where ordinary reinforcement and strands are likely to be in electrical contact, the potential of the strands themselves cannot be measured. Once strands are cathodically polarized, the corrosion probability for a given chloride content may be lower than that projected by the Markeset model.

In this study, exposing the reinforcement in location P1 (Figure 5b) confirmed the indicated 100% corrosion probability of the strands. However, for chloride concentrations of 0.344% by weight of concrete (1.81% by weight of cement) only small pitting was detected, whereas heavier corrosion would have been expected based on corrosion bridge studies [62]; see Figure 16. Moreover, for the high chloride content at the level of both strands and stirrups in location P1 (Table 3), stirrups exhibit more severe and uniform corrosion; see Figure 5b. This means that relying on chloride concentrations alone may give unreliable information about not only the probability, but also the severity of corrosion in NIB girder strands.

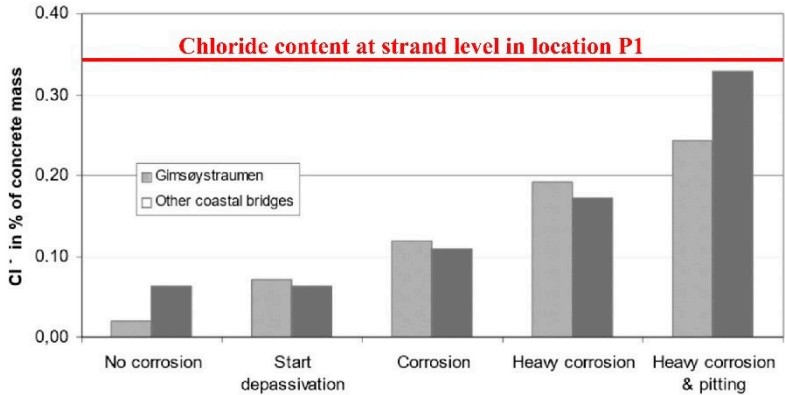

**Figure 16.** Relationships between corrosion severity and chloride content from more than 300 registrations on various coastal bridges in Norway [62]. Red line—chlorides at the strand level in location P1.

Because the bottom strands and stirrups in location P1 are electrically connected (although this was not tested), it is likely that this led to macrocell corrosion between the initially corroding ordinary reinforcement and the still passive strands. If so, both the chloride content required for corrosion initiation in the strands and the corrosion rate in the actively corroding reinforcement would have increased [22], leading to a considerably greater cross-section loss in the stirrups than in the strands. This is consistent with the observed state of the reinforcement. It may even be that the presence of the mounting bars and stirrups postponed the strand corrosion.

In addition, the poorer quality of the concrete–steel interface in the cracked cover of the stirrups and consequently higher oxygen and moisture supply could have enhanced the corrosion of stirrups. Moreover, differences in the production methods and chemical composition of the strands and ordinary bars could have had an impact on the propagation of their corrosion [63].

## 5.3. Combining NDT with DT Results

To analyze the correlation between NDT and DT results, the corrosion probabilities calculated for HCP based on statistical distributions in Figure 14 were compared with the probabilities calculated for chloride concentrations based on the Markeset model. The results are shown in Figure 17.

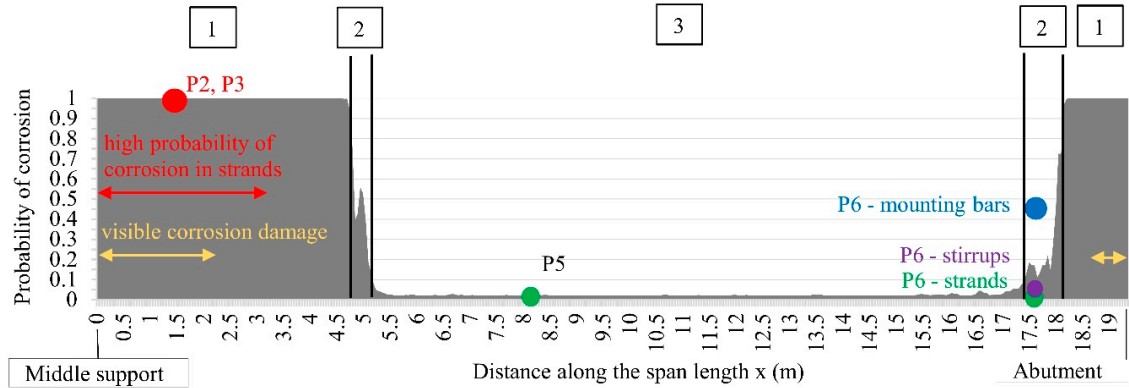

**Figure 17.** Comparison of calculated corrosion probability along Girder 2: grey area—corrosion probability based on HCP collected from support zones; dots—corrosion probability based on chloride concentrations; nos. 1, 2, 3—zones with high (>90%), uncertain, and low (<10%) corrosion probability, respectively.

The corrosion probabilities obtained from both kinds of test on Girder 2 are consistent for all reinforcements in locations P2, P3, and P5, which increases the reliability of assessment in the zones with the highest and lowest corrosion probabilities. Moreover, the zones with a corrosion probability greater

than 90% extend up to 2.5 m beyond the areas with visible corrosion damage; see Figure 17. This means that corrosion can be detected where there are no signs of corrosion visible on the concrete surface.

The uncertain corrosion probability estimated from HCP results in location P6 is related to mounting bars rather than strands; see Figure 17. Once the outermost reinforcement corrodes (mounting bars, stirrups), the HCP method may overestimate the corrosion probability of electrically connected strands. Accordingly, the area with a high corrosion probability in the strands may be less than that marked with grey in Figure 17. We can conclude that corrosion assessment based on the combined results from NDT (HCP, ER, visual assessment) and DT (chloride concentrations) can reliably identify only areas with different probabilities of reinforcement corrosion. Although strands are placed in concrete without ducts and the above approach is applicable for detecting their corrosion, the only way to confirm the condition of strands is to open the cover.

Nevertheless, a negligible corrosion probability was detected in location P5, where chloride content at the level of mounting bars was 0.04% by weight of concrete (0.20% by weight of cement). This means that the CEB [17] chloride threshold for prestressed structures was too conservative for the NIB girder in this study. In location P1, corrosion spots were found in the strands with a chloride content of 0.344% by weight of concrete, so corrosion in strands with a higher chloride level in location P2 is also to be expected; see Figure 18. Removal of cover in locations with severe damage allowed us to estimate the chloride threshold for initiation of strand corrosion. This was approximated at between 0.14 and 0.34% by weight of concrete; see Figure 18. Since strands probably corrode with potentials lower than the −370 mV CSE found in location P2, the area with a high probability of strand corrosion extends about 3.2 m from the middle support; see Figure 17.

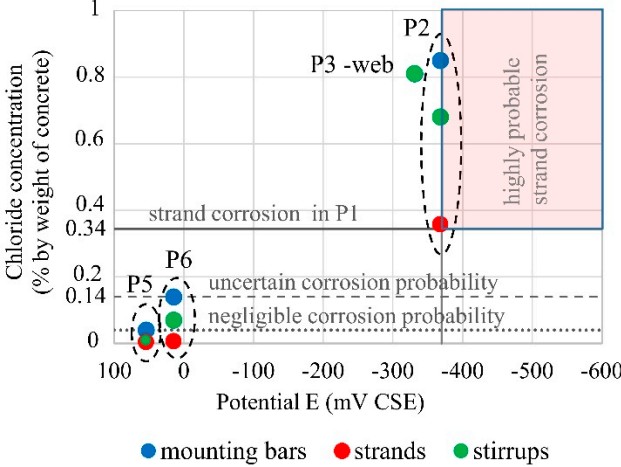

**Figure 18.** HCP and chloride concentrations for sampling locations in Girder 2. The chloride concentration at the level of the corroding strand in location P1 (in Girder 11) is included for comparison.

The high corrosion susceptibility of pretensioned girder support zones and their vicinities was found in our previous study [2]. The results of experimental tests in this study confirmed those findings.

## 6. Procedure for Inspection and Assessment of Pretensioned Girders

Based on our experience from this study, we recommend the following procedures for inspection and assessment of corrosion probability in pretensioned girders. The procedures integrate conventional test methods and are inspired by existing methodologies in the literature [22,45]. They include analyzing the HCP data from areas with comparable moisture conditions [48,49].

The critical girder(s) for detailed inspection should be based on a visual inspection of all of the bridge girders; see Figure 19. Considering findings in both our previous research [2] and this study, the 2nd and/or 3rd girder facing the sea are typically those with the highest probability of reinforcement corrosion damage due to chloride exposure.

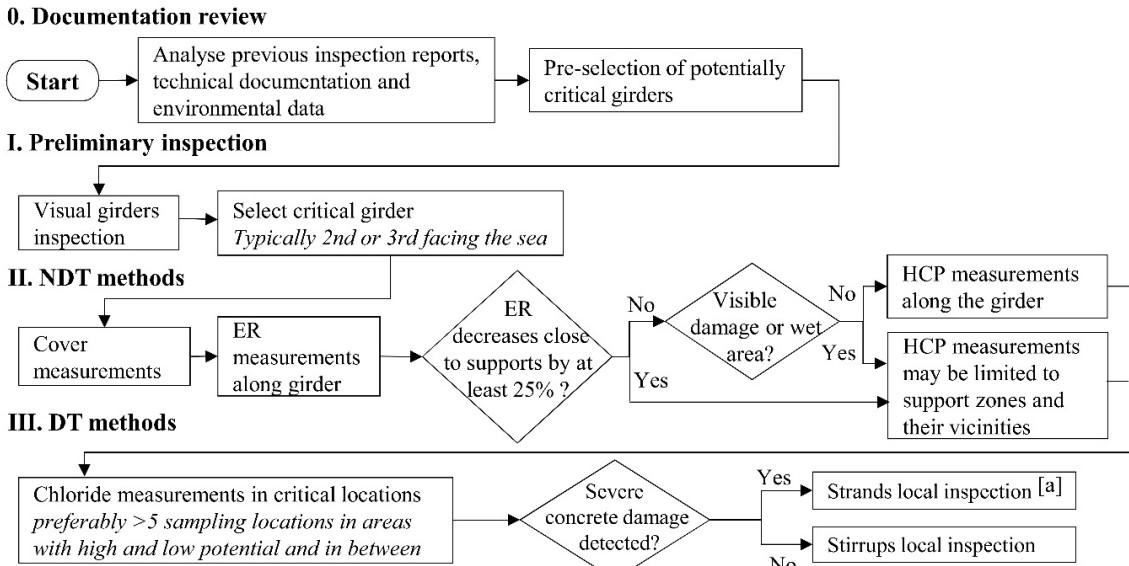

**Figure 19.** Procedure for detailed inspection of pretensioned bridge girders in a marine environment.

The ER should be measured first (recommended grid about 500 mm) and preliminarily analyzed. Because support zones and their vicinities are more susceptible to corrosion, the first sign of corrosion is likely to occur in this area. If ER readings decrease close to supports by more than 25% (the scatter expected in the field [23]) or corrosion damage or wet areas are detected, the HCP measurements can be limited to the support zones and their vicinities. This reduces the inspection time and cost. In this case, the HCP should be measured until consistently high HCP levels are detected over a distance of at least 2 m. Because a significant amount of data is required for statistical analysis, the HCP should be measured on a very small grid. We recommend 50 mm.

Samples for chloride concentration testing should be taken in at least five locations: where the HCP values are highest, lowest, and between these two locations. The corrosion condition can be confirmed by local inspection of the reinforcement in the cover zone. The removal of large amounts of concrete cover to inspect strands in their anchorage zone can considerably reduce their bond strength and impact their critical prestressing transfer and development length, which may increase the risk of strand slippage and should thus be avoided. Nevertheless, where severe concrete damage is detected, it is recommended that cover should be removed locally up to the strand level to confirm the corrosion condition for strands.

Data collection is described in Figure 19.

The assessment of the data collected is described in Figure 20. The statistical analysis of HCP should be performed following the procedure developed by Gulikers and Elsener [42] on data collected from areas where an increase in moisture is detected, e.g., support zones and their vicinities. Corrosion probabilities from measured chloride concentrations should be evaluated based on the statistical model for critical chloride distribution for similar structures and environmental conditions as proposed by Markeset [16]. The estimated corrosion probabilities from both chloride concentrations and HCP collected from areas with similar moisture conditions are then compared. Corrosion probability assessment can be considered reliable when both methods point to the same conclusions and coincide with the visual damage where present. The results can also be supported with reinforcement inspection results.

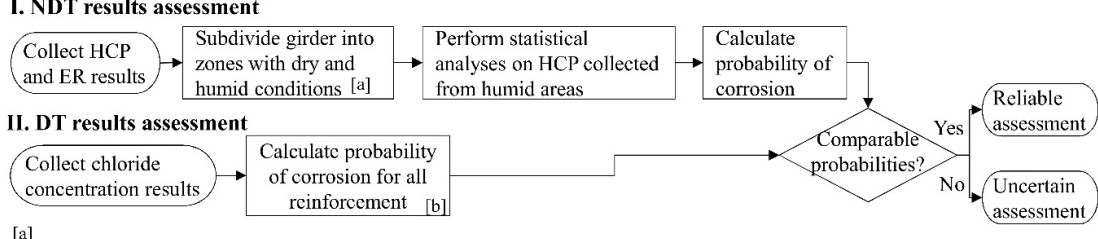

**I. NDT results assessment**

[a] Based on ER and HCP results together with visual observations of corrosion damage and/or wet consrete surface

[b] Based on statistical distribution of critical chloride content, i.e. Markeset model

**Figure 20.** Procedure for assessment of pretensioned bridge girders in a marine environment.

## 7. Conclusions and Recommendations

This experimental study on corrosion detection in pretensioned bridge girders exposed to the Norwegian coastal climate for 33 years showed that:

1.  Corrosion probability assessed on the basis of statistical analysis of HCP collected from areas with comparable moisture conditions was in good agreement with corrosion probability assessed on the basis of chloride concentrations. The combination of methods described in Section 6 enables reliable assessment of areas with varying corrosion probability. The zones with high corrosion probability extended beyond the visible corrosion damage, which indicates this approach can detect reinforcement corrosion before it is visible on the concrete surface. Although general areas with varying corrosion probabilities can be reliably estimated, the actual condition of the strands can only be found by removing concrete cover.

2.  The HCP method could not distinguish corrosion probability between closely spaced strands and other types of electrically connected reinforcement in the bottom flange of a girder (e.g., stirrups and mounting bars).

3.  Numerical criteria for corrosion probability assessment based on HCP given in ASTM C876 and for ER given in RILEM TC 154-EMC significantly underestimated areas with high corrosion probability in the case investigated. We cannot therefore recommend sole reliance on such absolute potential and resistivity thresholds in corrosion assessment. Moreover, we found HCP measurements to be more reliable than ER.

4.  Although very high chloride concentrations were measured at the level of strands (about nine times higher than the CEB conservative threshold of 0.2% by weight of cement) and severe concrete damage was observed around the ordinary reinforcement, only small corrosion spots were found on the strands. This could be due to macrocell corrosion delaying strand corrosion. Other factors could be differences in the quality of the concrete–steel interface and type of steel. Severe corrosion damage visible on the concrete surface does not necessarily indicate severe strand corrosion. High chloride concentrations alone cannot be interpreted as sufficient to indicate severe corrosion. Moreover, the assessment of corrosion probability from chloride concentrations alone could be uncertain due to the lack of research into the statistical distribution of chloride thresholds for prestressing reinforcement.

5.  We found corrosion in and near the support zones of NIB girders exposed to wetting–drying cycles (low ER, visible wet areas). In contrast, corrosion probability was negligible in the span where dry conditions were found (high ER). Higher moisture in combination with damaged or poor quality of concrete cover probably led to more than seven times higher maximum accumulated chloride concentration (up to 1.2% by weight of concrete) and consequently deeper chloride ingress in the support zones than in midspan.

6.  With regard to the 2nd girder from the side facing the sea, the exposure to moisture and chlorides near the supports was comparable for both the girder web and the bottom flange. A higher corrosion probability is also to be expected in the outer web surface facing the sea.

We recommend the procedure for assessing the overall probability of corrosion by integrating conventional test methods (HCP, ER, visual assessment, and chloride concentration measurement) and analyzing data from areas with similar exposure to moisture. We also recommend that inspection of NIB girders should focus more on the girder support zones, although the girder span should not be neglected. Where it is possible to limit measurements to the support zones, it can increase the time- and cost-effectiveness of the inspections.

Due to the lack of research into the statistical distribution for critical chloride thresholds for prestressing steel, we recommend further studies based on field data. We also recommend further study on the range of cathodic polarization (in relation to ER and anode/cathode size) resulting from macrocell corrosion between ordinary reinforcement and passive prestressing steel, in addition to the effect of electrochemical potential on the strand chloride threshold.

**Author Contributions:** Conceptualization, M.J.O., K.H., G.M., T.K., M.A.N.H.; methodology, M.J.O., K.H., and G.M.; validation, G.M., K.H., T.K., M.A.N.H.; formal analysis, M.J.O.; investigation, M.J.O., K.H., G.M.; data curation, M.J.O.; writing—original draft preparation, M.J.O.,K.H., G.M.; writing—review and editing, K.H., G.M., T.K. and M.A.N.H.; visualization, M.J.O., G.M, K.H., T.K., M.A.N.H.; supervision, G.M., T.K., M.A.N.H.; project administration, G.M., T.K.; funding acquisition, T.K., K.H., G.M. All authors have read and agreed to the published version of the manuscript.

**Funding:** This research was funded by Oslo Metropolitan University (OsloMET), Norwegian University of Sciences and Technology (NTNU) and Norwegian Public Road Administration (NPRA). The APC was funded by NTNU.

**Acknowledgments:** This study was supported by the 5-year research and development programme for bridges and quays "Bedre Bruvedlikehold" ("Better Bridge Maintenance") established by the Norwegian Public Road Administration (NPRA) in 2017. Financial support for the bridge investigation from NPRA's Northern Region is gratefully acknowledged. The authors also wish to thank Lise Bathen and Multiconsult for their technical support during inspection.

**Conflicts of Interest:** The authors declare no conflict of interest.

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
