# Peer review of "Inspection and Assessment of Corrosion in Pretensioned Concrete Bridge Girders Exposed to Coastal Climate"

_infrastructures, doi:10.3390/infrastructures5090076_

Round 1
Reviewer 1 Report
The work presents abundant results base on well-established methodologies (visual inspection, measurement of concrete electrical resistivity or half-cell potential, chloride threshold determination, and so on). Authors make an important effort to combine all of these methodologies. Based on the obtained results, they propose a procedure to assess the probability of corrosion of pre-tensioned concrete structures.
I think the paper is well-structured and experimental results well explained, therefore under my point of view, it is adequate to be published in the present form.
Author Response
We would like to thank very much the Reviewer for analysing and evaluating the article. We also thank for the positive opinion about the article content and structure.
Reviewer 2 Report
Line 40 – Aging is not a cause for corrosion by itself.
Line 44 – provide some general information for Norwegian marine environment (i.e. level of chloride, average temperature,
and humidity, etc.)
Line 49 – why corrosion was not observed on the surface of the first girders?
Line 56 – It also needs to be included that strands experience high levels of stresses during the bridge’s service life which increases the rate and likelihood of corrosion.
Line 71 – as this article describes the study that was done on a bridge, provide some sketches and information to better illustrates the condition such as a map view, girder and abutment layout, etc.
Line 75 – Chloride ingress also depends on the level of chloride the bridge is exposed to.
Author Response
We would like to thank very much the Reviewer for checking, evaluating, and commenting on the article. The comments helped to improve the article. The responses to the comments with detailed explanation are presented below.
1. Line 40 – Aging is not a cause for corrosion by itself.
Response:The text was corrected according to the above comment. Since corrosion is typically found in older structures, the text ‘’corrosion in pretensioned concrete bridge girders..’’ has been replaced with ‘’corrosion in aging pretensioned concrete bridge girders..’’ (Line 40).
2. Line 44 – provide some general information for Norwegian marine environment (i.e. level of chloride, average temperature, and humidity, etc.)
Response: The ‘’Norwegian marine environment’’ was replaced with the more correct and descriptive term ‘’Norwegian coastal climate’’. To explain the varying coastal climate in Norway, the general classification of the Norwegian coastal climate into 3 coastal climate zones (inner coastal, coastal, and harsh coastal) was added in the text in Line 43 and 44. This classification is based on the documents from the Norwegian Public Roads Administration and was presented in the paper and referred to as [2].
Due to geographical conditions, average temperature between different parts of Norway (for example south and north) may vary significantly. Consequently, the temperature and humidity averaged for the all Norwegian coastal climate zones may not be representative for the investigated bridge. However, to give an overview and complete the environmental data, the information about average relative humidity and temperature for the location of the investigated Dalselv Bridge has been added in Section 3.1. Bridge details, Line 224 and 225.
3. Line 49 – why corrosion was not observed on the surface of the first girders?
Response: In the study we are referring to ([2] Osmolska, M.J.; Kanstad, T.; Hendriks, M.A.N.; Hornbostel, K.; Markeset, G. Durability of pretensioned concrete girders in coastal climate bridges: Basis for better maintenance and future design. Struct. Concr. 2019, 20, 2256–2271.) corrosion was also observed on the surface of the first girders. However, compared to the inner girders (typically the second and third girders from the windward side, which is typically the side facing the sea), corrosion damage of the first girders (outermost) was less severe. Therefore, the location of the most severe corrosion damage is highlighted in the Introduction of this paper. The reference to [2] has been repeated for clarity (Line 50)
4. Line 56 – It also needs to be included that strands experience high levels of stresses during the bridge’s service life which increases the rate and likelihood of corrosion.
Response: This important information has been added to the text in Line 54-55. Few studies in the literature reported increase in the corrosion-induced mass loss of the strands for increasing stress level.
5. Line 71 – as this article describes the study that was done on a bridge, provide some sketches and information to better illustrates the condition such as a map view, girder and abutment layout, etc.
Response: The overview of the bridge layout (plan view of the girders and supports including abutments) is collected in Section 3.1. Bridge details together with the main information regarding girders design and environmental conditions. The girders and abutment layout are shown in the Figure 1b and Figure 1c (previously a and b). In addition, a photo was included to show the actual view of the bridge (Figure 1d (previously c)).
To illustrate the environmental condition/location a map view has been added in Figure 1a.
6. Line 75 – Chloride ingress also depends on the level of chloride the bridge is exposed to.
Response: The chloride ingress indeed depends on the so called ‘’chloride loads’’. The text has been improved according to the above comment. This information has been added in Line 78-79.